# Sodium-Doped 3-Amino-4-hydroxybenzoic Acid: Rediscovered Matrix for Direct MALDI Glycotyping of *O*-Linked Glycopeptides and Intact Mucins

**DOI:** 10.3390/ijms242316836

**Published:** 2023-11-28

**Authors:** Shogo Urakami, Hiroshi Hinou

**Affiliations:** 1Graduate School of Life Science, Hokkaido University, Sapporo 001-0021, Japan; urakamishogo27@eis.hokudai.ac.jp; 2Frontier Research Center for Advanced Material and Life Science, Faculty of Advanced Life Science, Hokkaido University, Sapporo 001-0021, Japan

**Keywords:** 3-amino-4-hydroxybenzoic acid, MALDI, glycotyping, *O*-glycan, mucin

## Abstract

3-Amino-4-hydroxybenzoic acid (AHB) was the first matrix identified by glycoprotein glycan analysis using matrix-assisted laser desorption/ionization mass spectrometry (MALDI-MS). However, compared to commonly used matrices, such as 2,5-dihydroxybenzoic acid (DHB), AHB is less efficient at glycan ionization and lacks the ability to ionize other molecular species, such as peptides, and thus is no longer used. In this study, we focused on the glycan-selective ionization ability of AHB and its low-noise properties in the low-molecular-weight region, as we expected that these properties could be enhanced by adding sodium to AHB. Sodium-doped AHB (AHB/Na) selectively imparts sodium adduct ions onto *O*-glycan fragments generated by the in-source decay (ISD) of glycopeptides and glycoproteins containing *O*-glycans that occurs during intense laser irradiation, enabling direct *O*-glycan analysis. Furthermore, we demonstrated that it is possible to investigate the internal structure of each *O*-glycan fragment with pseudo-MS/MS/MS using the sodium adduct ion of the *O*-glycan-derived ISD fragments from an intact mucin mixture.

## 1. Introduction

Protein glycosylation plays an important role in various biological processes, including cell-to-cell signal transduction, infection, and disease [1]. Glycosylation disorders have been experimentally utilized to predict and diagnose diseases using carbohydrate antigens (CAs) [2]. Based on amino acid differences, mammalian protein glycosylation is classified based on *N*-glycans and *O*-glycans. In particular, *O*-glycans are widely used as clinical CAs because they are diverse near protein binding sites and significantly affect peptide conformation [3,4,5]. Furthermore, some *O*-glycans are involved in the survival of polar fish, as they confer antifreeze activity through conformational control of the peptide backbone [6,7].

Matrix-assisted laser desorption/ionization (MALDI) is one of the soft ionization methods utilized with the time-of-flight mass spectrometry (TOFMS) of macromolecules, such as biopolymers [8,9]. In MALDI-TOFMS, choosing a matrix that is mixed with the analyte is critical, and many suitable matrices for various analytes, such as proteins, lipids, and glycans, have been reported [10,11]. In the MALDI-TOFMS of glycans, 3-amino-4-hydroxybenzoic acid (AHB; Figure 1) was the first reported matrix to detect free glycans released from glycoproteins [12]. However, AHB was soon replaced by 2,5-dihydroxybenzoic acid (DHB; Figure 1) [13,14], which was discovered in the same year (1991) due to its high ionization efficiency and versatility. Nowadays, DHB is the gold standard matrix for glycan analysis, and few studies utilize AHB [15,16].

In general, the MALDI-TOFMS of glycoprotein glycans involves the cleavage and separation of the glycan component from the protein, which is then mixed with the matrix for analysis [15]. The addition of aniline derivatives to DHB has been reported to improve the ionization efficiency of cleaved glycan components [17,18]. Recently, we reported that MALDI in-source decay (ISD) allows for the direct TOFMS of *N*-glycan structures by simply mixing glycoproteins, an aniline-type base, and alkaline-metal-doped DHB (aniline/DHB/Na or 1,5-diaminonaphthalene (DAN; Figure 1)/DHB/Na) [19,20]. However, matrix-derived noise has made it difficult to directly analyze *O*-glycans on glycoproteins that have a molecular weight of less than 1000 using these organic salt-type matrices in MALDI-ISD TOFMS.

Here, we report that AHB/Na (sodium-doped AHB) is suitable for use in the MALDI-ISD TOFMS of *O*-glycans in intact glycopeptides and glycoproteins. AHB/Na minimizes matrix-derived noise and enables sharp *O*-glycan-derived ion peaks in the low-molecular-weight region, allowing for the direct analysis of the *O*-glycan pattern of the analytes using MALDI-ISD MS. In addition, pseudo-MS^3^ analysis [21,22] of the glycan-derived ISD ion peak also enables the identification of the constituent monosaccharide residues. This ultra-rapid method for *O*-glycan analysis using AHB/Na was investigated to pave the way for practical MALDI glycotyping [20] of mucus/glycocalyx and its medical application.

## 2. Results and Discussion

### 2.1. MALDI-ISD Analysis of O-Linked Cyclic Glycopeptides

Cyclic antifreeze glycopeptides (cAFGPs; Figure 2a) [23], which have a cyclic peptide skeleton composed of a tripeptide unit with a core 1 disaccharide sidechain, were selected as model molecules in this study to identify suitable matrices for direct analysis of glycoprotein *O*-glycans using MALDI-ISD TOFMS. We tested four MALDI matrices, DAN, DHB (0.1% TFA), DAN/DHB/Na, and AHB/Na, for use in the MALDI-ISD TOFMS analysis of cyclic AFGPs. To promote ISD fragmentation of the analyte, laser irradiation power for the MALDI-TOFMS analysis was fixed at 80%. To the best of our knowledge, DAN [24,25] and DAN/DHB/Na [20] are suitable matrices for the MALDI-ISD analysis of the peptide region and *N*-glycan region from intact glycoproteins, respectively. Figure 2 shows the measurement results of cAFGPs for the four matrices. In all matrix systems, cAFGP-derived peaks were detected as sodium adduct ions, [M + Na]^+^, regardless of the addition of sodium ions. The use of the DAN matrix allowed for the detection of cAFGPs at a mass-to-charge ratio (*m*/*z)* 1240 (*n* = 2), 1848 (*n* = 3), and 2456 (*n* = 4) without its fragment peaks (Figure 2b). This result suggests that DAN is unsuitable for use in the ISD analysis of cAFGPs. On the other hand, the use of DHB (Figure 2c), DAN/DHB/Na (Figure 2d), and AHB/Na (Figure 2e) led to the detection of signals corresponding to both intact cAFGPs (*m*/*z* 1240, 1848, 2456, 3064, and 3673) and its ISD fragment peaks (*m*/*z* 1465, 1483, and 1686), which corresponds to the neutral loss of [hexose (Hex) + *N*-acetyl hexosamine (HexNAc) + H_2_O], [Hex + HexNAc], and [Hex] from the intact cAFGPs (*m*/*z* 1848, with a cyclic nonapeptide skeleton), respectively. Interestingly, the DAN/DHB/Na and AHB/Na matrices give higher ISD peaks corresponding to the neutral loss of [Hex + HexNAc + H_2_O] (*m*/*z* 1465 and 2073) rather than the neutral loss of [Hex + HexNAc] (*m*/*z* 1483 and 2091) from the cAFGPs. This indicates these two matrices promote the release of a free *O*-glycan moiety due to β-elimination-type cleavage at the threonine side chain during the ISD fragmentation process.

We next focused on forming ISD fragment ions corresponding to the released core 1 disaccharide moiety observed in the low-molecular-weight region (Figure 3). No ISD fragment peaks corresponding to core 1 disaccharides were observed when the DAN matrix was used (Figure 3a). In contrast, when the DHB, DAN/DHB/Na, and AHB/Na matrices were used, the sodium adduct peaks corresponding to the core 1 disaccharide fragment ion signals at an *m*/*z* 388 for [HexNAc + Hex + Na]^+^ and an *m*/*z* 406 for [HexNAc + Hex + H_2_O + Na]^+^ were observed, respectively (Figure 3b–d, and Appendix A). Similar to the results in Figure 2, where neutral losses of *O*-glycan disaccharides from cAFGPs were observed, free core 1 disaccharide-derived sodium adduct ions at an *m*/*z* 406 that were formed through the β-elimination from the threonine side chain were preferentially observed with the DAN/DHB/Na and AHB/Na matrices. Interestingly, in a previously reported direct MALDI-ISD analysis of *N*-linked glycoproteins, A ion pairs (^0,2^A and ^2,4^A; Δ*m*/*z* 60) derived from *N*-glycans occurred preferentially [19,20]. In contrast, in this study’s MALDI-ISD analysis of mucin-type glycopeptides, B and C ion pairs (Δ*m*/*z* 18) derived from *O*-glycans were observed.

Of the four tested matrices, AHB/Na clearly gave the narrowest (highest-resolution) peak signal (Figure 3d). The signal-to-noise (S/N) ratio of the free core 1 disaccharide signal at an *m*/*z* 406, which was formed due to the ISD of cAFGPs using a sodium adduct ion with DHB, DAN/DHB/Na, and AHB/Na matrices, was 38.7 (Figure 3b), 79.2 (Figure 3c), and 422.6 (Figure 3d), respectively. In addition to glycan-selective ionization efficiency [12], nucleophilicity, the suppression of total ion production during intense laser irradiation, and morphological homogeneity(Figure 3e and Appendix A) [19,20] are expected to contribute to the high resolution and high S/N ratio of *O*-glycan-derived ISD ions produced when using this AHB/Na matrix. The properties of the AHB/Na matrix, which gives ISD fragment ions with a significantly higher S/N ratio in the low-molecular-weight region, are expected to be suitable for the measurement of *O*-glycan fragments whose signals arise from the low-molecular-weight region. Next, we investigated whether the *O*-glycan structure could be directly observed by MALDI-ISD analysis of natural mucins.

### 2.2. MALDI-ISD O-Glycan Analysis of Intact Mucin

Mucins are representative of highly *O*-glycosylated proteins that build the gel layer that covers the mucosal surfaces of the body and serve as the primary barrier for permeant selection [26]. We selected porcine stomach mucin (PSM), one of the in vivo mucins that acts as a multi-mucin complex to protect the gastric secretory cell layer [27], to investigate the effect of the matrix on the direct analysis of *O*-glycans. As with the cAFGPs, PSM was mixed with four matrices (DAN, DHB, DAN/DHB/Na, and AHB/Na) without pretreatment and subjected to MALDI-ISD analysis. Unlike the cAFGP analysis, no peptide nor glycopeptide-derived PSM fragment signals were detected with the matrixes tested. As with the cAFGPs, no *O*-glycan-derived fragment signals were detected with the DAN matrix (Appendix A). When DHB or DAN/DHB/Na were used as matrices, faint signals corresponding to *O*-glycan fragments were observed in the matrix-derived strong noise signals (Appendix A, Appendix A). DAN/DHB/Na showed a stronger *O*-glycan signal from PSM than DHB, but the matrix-derived noise was also strong in the low-molecular-weight region, making this matrix unsuitable for the direct observation of mucin *O*-glycans (Appendix A). Notably, when PSM was mixed with the AHB/Na matrix, ISD fragment peaks corresponding to *O*-glycan fragments were attributed to the mono- and octasaccharide regions. Despite matrix-derived dense noise peaks in the low-molecular-weight region, the significantly high resolution and S/N ratio associated with using the AHB/Na matrix allowed for the attribution of glycan-derived fragments due to characteristic mass differences, indicating an increase or decrease in sugar residues (Appendix A). Furthermore, a paired signal with an Δ*m*/*z* 18, corresponding to the difference between free glycan fragments produced by β-elimination (C ions) and those produced by the cleavage of glycosidic bonds (B ions), supports the attribution of *O*-glycan-derived ions. Furthermore, C ion fragments were preferentially observed in the larger glycan regions (Figure 4, Appendix A).

The *O*-glycan-derived C ions observed in the MALDI-ISD MS measurements of the AHB/Na matrix and PSM mixture are in good agreement with the reported MALDI-MS spectra detected after separation and purification of free *O*-glycans released during a base treatment of PSM [28,29]. However, the C ions shown in Figure 4 would be expected to be observed even when free *O*-glycans hydrolyzed from PSM are mixed in the sample. A mixture of PSM and the benzyloxyamine (BOA)/DHB/Na matrix was also analyzed to demonstrate that the ISD of *O*-glycans on PSM generated the C ions. As previously reported, the mixture was stored at 60 °C for one hour after mixing with the samples to ensure that the BOA entirely tagged the reducing ends of the free glycans in the samples [28]. However, as shown in Appendix A, no signals corresponding to *O*-glycans tagged by BOA were observed in the mixture of the BOA/DHB/Na matrix and PSM. This indicates that all the *O*-glycan-derived signals observed in the mixture of PSM with the AHB/Na matrix were caused by the ISD of PSM. N: *N*-acetyl hexosamine; H: hexose, F: fucose (deoxyhexose).

### 2.3. Pseudo MS^3^ O-Glycan Analysis of cAFGPs and Intact Mucin with the AHB/Na Matrix

Next, the major signals from the *O*-glycan-derived ISD ions observed with the AHB/Na matrix were selected as precursor ions for pseudo-MS^3^ analysis using MALDI-LIFT-TOF/TOF [19,20]. First, the sodium adduct ion of the free core 1 disaccharide at an *m*/*z* 406, which was observed in both cAFGPs and PSM, was selected as the precursor ion. An almost identical fragment pattern of the core 1 disaccharide was observed from the precursor ion formed through ISD in each analyte (Figure 5, Appendix A). The precursor ion of this core 1 disaccharide has a reducing terminal hydroxyl group attached to the *N*-acetylhexosamine residue, whereas in the TOF/TOF spectrum, the fragments corresponding to the free hexoses and the *N*-acetylhexosamine residue that lost the reducing terminal hydroxyl group were observed as the sodium adduct ions [Hex + H_2_O + Na]^+^ and [HexNAc + Na]^+^, respectively.

In the pseudo-MS^3^ analysis of PSM with the AHB/Na matrix, the major *O*-glycan-derived ISD precursor ions corresponding to disaccharides and hexasaccharides (*m*/*z* of 388, 429, 534, 552, 568, 591, 609, 737, 771, 794, 899, 940, 1102, and 1120) yielded signal patterns indicative of sugar composition (Figure 6 and Appendix A). Figure 6 shows the pseudo-MS^3^ spectrum of ISD precursor ions corresponding to a B ion (*m*/*z* 1102) and C ion (*m*/*z* 1120) pair with a hexasaccharide composed of one deoxyhexose, two hexoses, and three *N*-acetylhexosamines. A comparison of fragment patterns in both spectra shows that B-ion fragments are preferentially produced from B-derived precursor ions, whereas fragments of both B and C ions are observed as coming from C-derived precursor ions. This fragmentation pattern derived from precursor ion pairs with the same glycan composition is expected to contain detailed glycan structural information. Since glycans with free-reducing ends are prone to fragmentation, further pretreatments, such as labeling and reduction, have commonly been used for the MS analysis of released glycans from glycoproteins [15,16]. Therefore, structural analysis studies based on the comparison of fragmentation patterns from *O*-glycan-derived B and C ions as precursor ions have been limited. As the PSM used in this study is a mixture of multiple mucins and is expected to have structural diversity even within the same glycan composition, a comparison with mucin-type glycopeptides with well-defined glycan structures, such as cAFGPs, will lead to a more detailed structural discussion.

## 3. Materials and Methods

### 3.1. Materials and Reagents

Type III mucin from porcine stomach (PSM) and 1,5-diaminonaphthalene (DAN) were purchased from Sigma-Aldrich Corp. (St. Louis, MO, USA). DHB, sodium bicarbonate (NaHCO_3_), O-benzylhydroxylamine (BOA), and acetonitrile (HPLC grade) were purchased from Wako Pure Chemical Industries, Ltd. (Osaka, Japan). 2,2,2-Trifluoroacetic acid (TFA) was purchased from Watanabe Chemical Industry Co., Ltd. (Hiroshima, Japan). 3-Amino-4-hydroxybenzoic acid (AHB) was purchased from Tokyo Chemical Industry (Tokyo, Japan). Additionally, 700 µm and 900 µm STA µFocus plates measuring 24 × 16 c were purchased from Hudson Surface Technology (Fort Lee, NJ, USA).

### 3.2. Matrices

Solutions of 50 mM DAN in acetonitrile/H_2_O (1:1, *v*/*v*), 50 mM AHB in acetonitrile/H_2_O (1:1, *v*/*v*), 600 mM BOA in acetonitrile/H_2_O (1:1, *v*/*v*), 500 mM DHB in acetonitrile/H_2_O (9:1, *v*/*v*), and 100 mM NaHCO_3_ in H_2_O were prepared. The 50 mM AHB was vortexed and sonicated. The DAN/DHB/Na matrix was prepared by mixing 4 μL of 50 mM DAN, 2 μL of 500 mM DHB, and 1 μL of 100 mM NaHCO_3_ and diluting it to 100 μL with acetonitrile/H_2_O (1:1, *v*/*v*). The AHB/Na matrix was prepared by mixing 20 μL of 50 mM AHB and 1 μL of 100 mM NaHCO_3_ and diluting it to 100 μL with acetonitrile/H_2_O (1:1, *v*/*v*). The DAN matrix was prepared by mixing 20 μL of 50 mM DAN and diluting it to 100 μL with acetonitrile/H_2_O (1:1, *v*/*v*). The DHB matrix was prepared by mixing 2 μL of 500 mM DHB and diluting it to 100 μL with acetonitrile/H_2_O/TFA (50:50:0.1, *v*/*v*/*v*). The BOA/DHB/Na matrix was prepared by mixing 2 μL of 600 mM BOA, 2 μL of 500 mM DHB, and 1 μL of 100 mM NaHCO_3_ and diluting it to 100 μL with acetonitrile/H_2_O (1:1, *v*/*v*).

### 3.3. Sample Preparation for MALDI TOF MS Analysis

Matrix solutions (0.35 µL) were spotted on the 700 µm or 900 µm STA µFocus MALDI plate (Hudson Surface Technology, Fort Lee, NJ, USA), and then analytes of 1 g/L (0.35 µL) were deposited on the pre-spotted plates. The plates spotted with the DAN/DHB/Na, AHB Na, DAN, and DHB matrices were dried at 25 °C. The plates spotted with the BOA/DHB/Na matrix were dried in a thermostatic chamber for 1 h at 60 °C. The morphological image of each matrix on the µFocus plate was taken by polarized camera DZK 33UX250 (The Imaging Source, Charlotte, NC, USA) with polarized filter equipped light source LG-PS2 (Olympus, Tokyo, Japan).

### 3.4. MALDI-TOF and TOF/TOF MS

All spectra were acquired using a Ultraflex III instrument (Bruker, Bremen, Germany) equipped with a 200 Hz Smartbeam Nd:YAG laser (355 nm). The ISD spectra from an intact glycopeptide and glycoprotein were acquired in the positive reflectron mode with 1000 laser shots, 80% laser power, and a random walk. The generated ions were accelerated to a kinetic energy of 25.0 kV. The TOF/TOF spectra were acquired with 2400 laser shots in the parent and fragment mode using a random walk. In the LIFT-TOF/TOF mode, the ISD fragment ions were accelerated to 8 kV in the MALDI ion source and selected within a time gate. The selected ISD ion was further accelerated to 19 kV in the LIFT cell. The metastable post-source decay (PSD) ions were analyzed without any additional fragmentation process, such as collision-induced dissociation.

## 4. Conclusions

In conclusion, AHB/Na allows for the glycan-selective MALDI-ISD analysis and pseudo-MS^3^ analysis of intact *O*-linked glycopeptides and glycoproteins without pretreatment. AHB/Na facilitated the release of free *O*-glycans through the β-elimination of the serine/threonine residue’s side chains; this gave *O*-glycan-derived ISD ion peaks with a high resolution and high S/N ratio due to the uniform morphology of the matrix, allowing for high glycan-selective ionization and suppression of total ion production during intense laser irradiation to promote ISD. The ISD products of *O*-linked glycans were formed as B and C ion pairs. In addition to synthetic *O*-linked glycopeptides (cAFGPs) with well-defined structures, we have shown that glycan patterns can be detected from mixtures, such as gastric mucins, composed of multiple *O*-linked glycoproteins without a pretreatment such as digestion or purification. Pseudo-MS^3^ analysis of the *O*-glycan-derived B- and C-type ISD ion pairs could provide more detailed structural information about the detected *O*-glycans. Matrices that selectively ionize peptide and *N*-glycan fragments from glycoproteins through MALDI ISD analysis have been reported. This study adds AHB/Na as a suitable matrix for the direct MALDI ISD analysis of *O*-glycans from glycoproteins. These ultra-fast approaches can significantly contribute to the quality control of glycoproteins in pharmaceuticals, the evaluation of biological samples, and glycomics research.

## Figures and Tables

**Figure 1 ijms-24-16836-f001:**
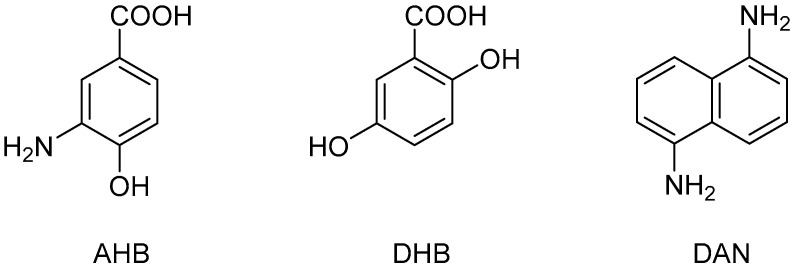
Chemical structure of the matrix used in this study.

**Figure 2 ijms-24-16836-f002:**
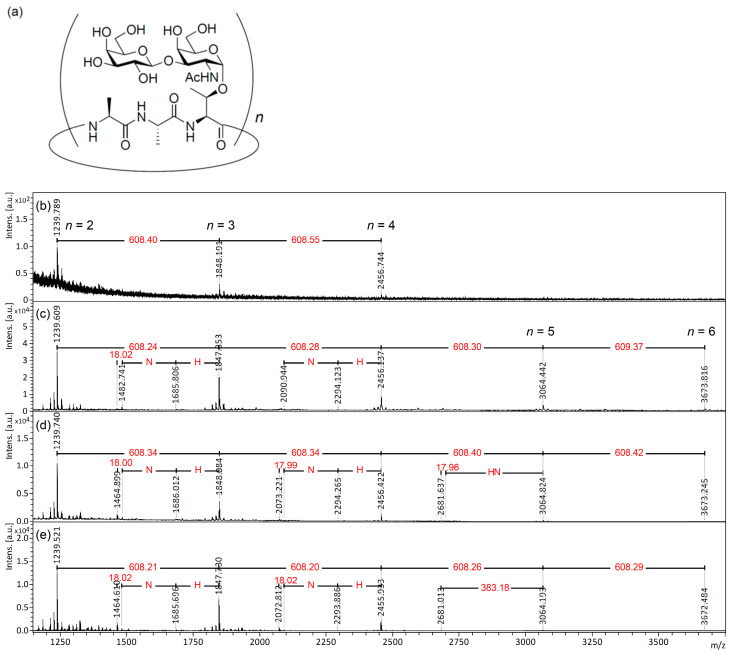
(**a**) Chemical structure of cAFGPs; (**b**–**e**) MALDI and MALDI-ISD mass spectrum of cAFGPs (1 μgμL^−1^) with the matrices: (**b**) DAN, (**c**) DHB, (**d**) DAN/DHB/Na, (**e**) AHB/Na. N: *N*-acetyl hexosamine; H: hexose.

**Figure 3 ijms-24-16836-f003:**
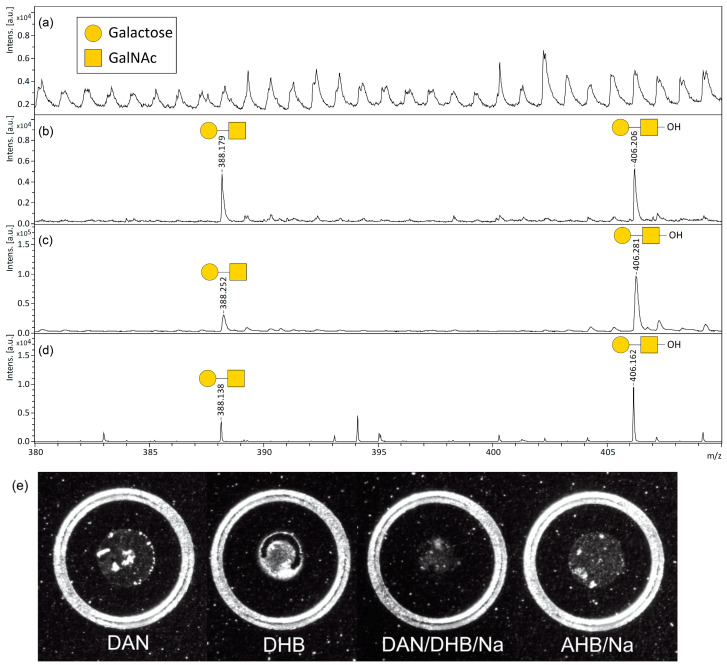
(**a**–**d**) MALDI and MALDI-ISD mass spectrum of cAFGPs (1 μgμL^−1^) in the range of core 1 disaccharide-derived sodium adduct ions with the following matrices: (**a**) DAN, (**b**) DHB, (**c**) DAN/DHB/Na, (**d**) AHB/Na. (**e**) On target plate morphology of each matrix used in this study. The outline of the circle is 2.9 mm.

**Figure 4 ijms-24-16836-f004:**
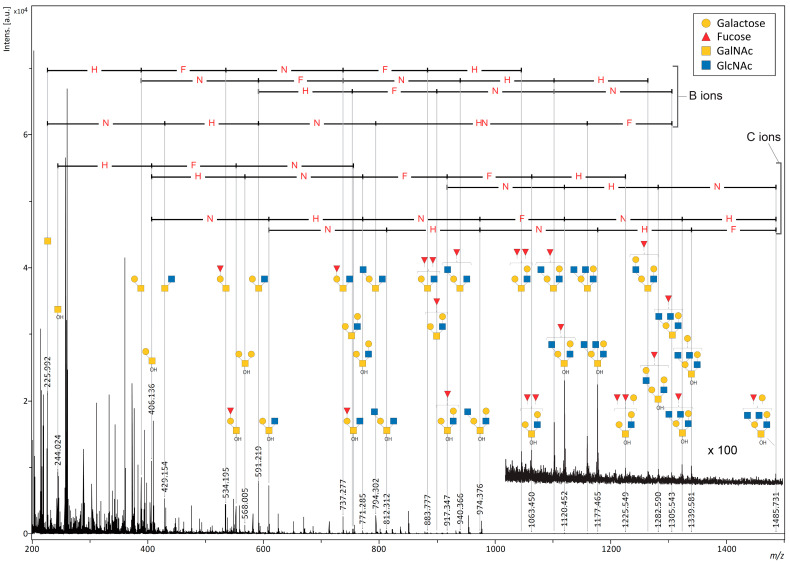
MALDI-MS spectra of *O*-glycan-derived sodium adduct ions produced by ISD fragmentation under intense laser irradiation of porcine stomach mucin (PSM) with AHB/Na matrix. N: *N*-acetyl hexosamine; H: hexose, F: fucose (deoxyhexose).

**Figure 5 ijms-24-16836-f005:**
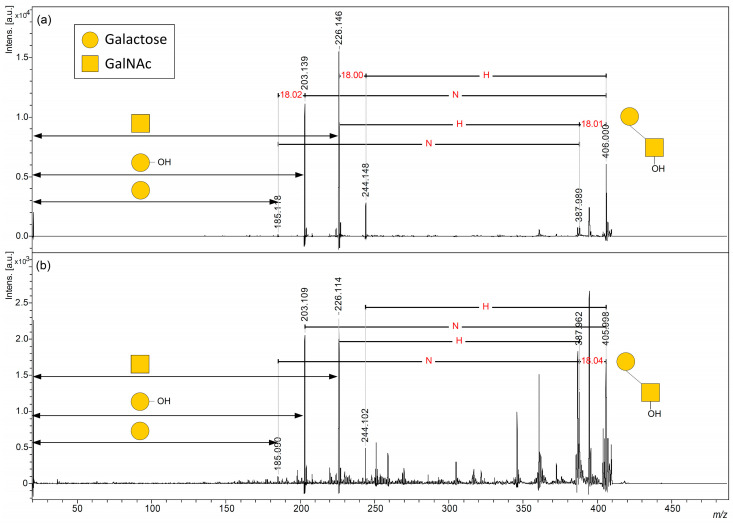
LIFT-TOF/TOF mass spectrum of core 1 disaccharide fragment [HexNAc + Hex + H_2_O + Na]^+^ at *m*/*z* 406 for precursor ion of (**a**) cAFGP (1 μgμL^−1^) and (**b**) PSM (1 μgμL^−1^) with AHB/Na. N: *N*-acetyl hexosamine; H: hexose.

**Figure 6 ijms-24-16836-f006:**
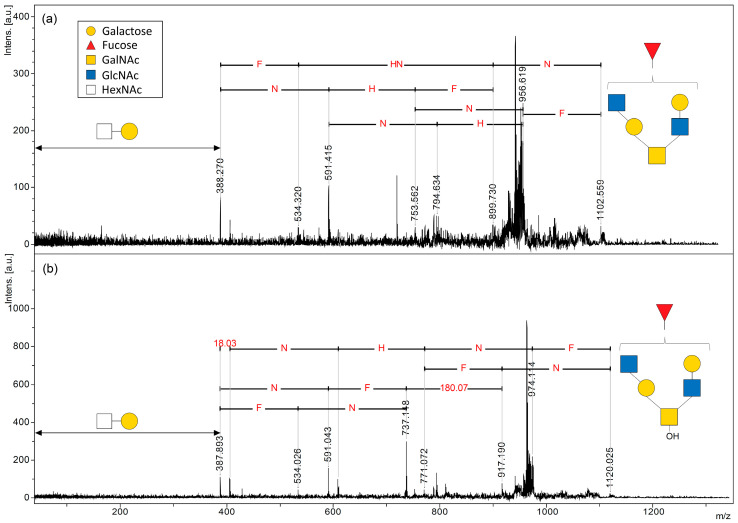
Pseudo-MS^3^ spectrum of PSM (1 μgμL^−1^) with AHB/Na (**a**) at *m*/*z* 1102 for precursor ion; (**b**) at *m*/*z* 1120 for precursor ion. N: *N*-acetyl hexosamine; H: hexose, F: fucose (deoxyhexose).

## Data Availability

All data generated or analyzed during this study are included in this published article and in the Appendix A.

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
