# Peer review of "Sodium-Doped 3-Amino-4-hydroxybenzoic Acid: Rediscovered Matrix for Direct MALDI Glycotyping of *O*-Linked Glycopeptides and Intact Mucins"

_ijms, 2023, doi:10.3390/ijms242316836_

Round 1
Reviewer 1 Report
Comments and Suggestions for Authors
The authors attempt to show that AHB, with the addition of sodium can be an effective matrix when used for MALDI glycan analysis.
The results show a good signal-t-noise ratio. Better than what's found with DHB-TFA, the most common matrix employed. The conclusion that sodium doped AHB is a suitable matrix is a reasonable one and supported by the results.
Author Response
Thank you for your positive feedback. We will develop this finding into practical mucus/glycocalyx MALDI glycotyping.
Reviewer 2 Report
Comments and Suggestions for Authors
Protein glycosylation plays a significant role in biological processes and is classified into N-glycans and O-glycans based on amino acid differences. To analyze glycans, authors employ the Matrix-assisted laser desorption/ionization (MALDI) time-of-flight mass spectrometry (TOFMS) method. In this study, authors used, several types of matrices, and among them 3-amino-4-hydroxybenzoic acid (AHB) matrix with sodium adducts was investigated for the direct analysis of O-glycans. This matrix minimizes noise and enables sharp ion peaks, allowing for a direct insight into low-molecular-weight O-glycans using the MALDI-ISD TOFMS method. Furthermore, pseudo-MS3 analysis of glycan-derived ion peaks facilitates the identification of monosaccharide residues, improving our understanding of protein glycosylation.
The manuscript is well-written, well-structured and easily readable. Additionally, the authors have paid attention to including all relevant references, notably acknowledging the historical contributions of two key groups of scientists in the development of the MALDI-TOF method: the Karas & Hillenkamp group and the group associated with Koichiro Tanaka, which is highly commendable.
The manuscript is adhering to the journal’s standards, however list of issues need to be addressed, and thus in my opinion - minor revision is required before it should be in consideration for publishing:
Introduction section:
Line 41: ''which was discovered the same year...''
Authors should add this year in text (1991)
While introduction section provide many information, authors need to more clearly define the purpose and objectives of the research. In other words, state more clearly what the aim with this research. This should be in the last paragraph of the Introduction section.
Materials and Methods section, subsection 3.2 Matrices
Line 216: why was it necessary to use 9:1 v/v ratio for acetonitrile/H2O solution for DHB?
General:
Is reference [28] necessary to include? Don't the other references already cover all the claims mentioned in the text?
Author Response
We sincerely appreciate your positive and important suggestions for improvements to our manuscript.
We have made the following revisions following your suggestions.
Introduction section
line 41: We have stated the year in which both matrices were discovered.
"which was discovered the same year (1991),"
In the last paragraph of the Introduction section: We have clearly stated the purpose of this study as follows.
"This ultra-rapid method for O-glycan analysis using AHB/Na was investigated to pave the way for practical MALDI glycotyping [20] of mucus/glycocalyx and its medical application."
Materials and Methods section, subsection 3.2 Matrices
Line 216: The reason for using acetonitrile/H2O in a 9:1 ratio was to dissolve a high concentration (500 mM) of DHB, which was diluted 50-fold with a 1:1 acetonitrile/H2O solvent to a final concentration of 10 mM.
General: The literature [28] reports a matrix [benzyloxyamine (BOA)/DHB/Na] that can label the reducing ends of free glycans by oxime formation. Since this study uses this matrix to demonstrate the absence of free O-glycans in porcine gastric mucin (PSM), a citation to the literature [28] is necessary.
However, p5line197 does not require a citation to this reference [28]. Reference [28] has been removed from this p5line197 citation.
Reviewer 3 Report
Comments and Suggestions for Authors
This paper re-visits the use of 3-amino-4-hydroxybenzoic acid (AHB), the original but discarded MALDI matrix, with added sodium to analyse O-glycans from porcine stomach mucin. The matrix is used because of the production of only low abundance ions in the low molecular mass region, thus making it a suitable matrix for analysis of low molecular mass compounds. The matrix is found to produce much sharper ion peaks than the widely used matrix 2,5-dihydroxybenzoic acid and various mixtures of these two matrices and a third matrix (1,5-diaminonaphthalene (DAN)). When applied to a glycopeptide and the mucin samples, prominent in-source decay fragment ions are produced consisting essentially of glycosidic cleavages.
The experiments reported in this paper are designed to produce ISD fragment ions by use of a high laser power. From an analytical point of view, the initial MS spectrum should be as devoid of fragment ions as possible so that it reflects the composition of the sample. C-Type fragments, as are produced in these experiments, are isobaric with native glycans and their presence can confuse analytical results. Spectra should initially be recorded at a lower laser power to maximize the relative abundance of the molecular ions.
Analysis of the glycopeptide sample (a mixture of glycosylated cyclic peptides) is shown in Figure 2 with the four matrices tested. The most prominent molecular ions are produced from DHB, not the AHB matrix. AHB also produces more prominent fragment ions, including C-type ions which could be mistaken for additional compounds.
Figure 3d (spectrum of Gal-GalNAc) – This spectrum contains several unlabelled ions (e.g. m/z 394) not present in the other spectra in this figure. What are they? The ion peaks in this spectrum are sharper than those from the other matrices. Have the authors ruled out the possibility that this effect could be caused by the use of different focusing conditions of the instrument?
Figure 3 and later – The symbols used for the glycans should be defined. Do the angles connecting the symbols represent linkage?
Figure 3(e) The photographs show the MALDI targets produced by each matrix but there is no discussion in the text.
The analysis of the mucin sample presents a similar problem to that of the glycopeptide. Here, only fragment ions and no molecular ions are observed and so it is not possible to differentiate ions corresponding to intact glycans from C fragments produced from the larger glycans. Thus, the composition of the sample cannot be deduced from the spectrum. Each ion is then fragmented but only B and C glycosidic fragments appear to be formed. These ions are insufficient to determine the detailed structure of the ions (i.e. no cross-ring cleavages are reported). Yet, Figure 4 shows detailed structures, including the linkage between the monosaccharide constituents. How were the structures of these ions obtained? Many of the ions shown in the fragmentation spectra are unlabelled. Can the authors comment on the nature of these ions?
Figure 5 - the spectrum (b) contains many prominent ions not present in spectrum (a) even though the ions selected for fragmentation are nominally identical. Can the authors comment on the nature of these additional ions?
Supplementary figures S5, S6: Spectra show, in panels (b) contain ions which appear to be those from the matrix and to be present at abundancies similar to those from the sample. Surely, this is a major disadvantage?
Figures S8, S9 – most of the ions are unlabelled and many appear to be from the matrix. The labelled glycan-derived ions appear to be of very low abundance. Figures S10b to S19b – There is no indication as to what the ions represent.
References
Some points need attention:
Some journal titles are abbreviated, others are not.
Reference 9 – Normal text should be used for the article title, not upper case.
Reference 15 - The volume and page numbers are missing.
Reference 16 – The volume number is missing.
In conclusion, the problem with this work is that there is no way to determine the glycan composition of the mucin sample because the liberated glycans have similar masses to C-type fragments that can be formed from many of the glycans. The authors should investigate conditions for producing molecular ions from the glycans in the mucin sample other than by the production of ISD fragments. Possibly this will mean chemically liberating the glycans. Then the AHB matrix should be compared with the other matrices for their ability to produce molecular ion. Following this, fragmentation can be performed on the molecular ions known to be from native glycans, possibly by a method that produces cross-ring fragments in addition to the reported glycosidic fragments.
Author Response
First, we would like to thank you sincerely for reviewing our manuscript.
Reviewer Comment: This paper re-visits the use of 3-amino-4-hydroxybenzoic acid (AHB), the original but discarded MALDI matrix, with added sodium to analyse O-glycans from porcine stomach mucin. The matrix is used because of the production of only low abundance ions in the low molecular mass region, thus making it a suitable matrix for analysis of low molecular mass compounds. The matrix is found to produce much sharper ion peaks than the widely used matrix 2,5-dihydroxybenzoic acid and various mixtures of these two matrices and a third matrix (1,5-diaminonaphthalene (DAN)). When applied to a glycopeptide and the mucin samples, prominent in-source decay fragment ions are produced consisting essentially of glycosidic cleavages.
Author’s reply: First, we would like to thank you sincerely for reviewing our manuscript.
As you mentioned, this manuscript reports the formation and characterization of pronounced in-source fragment ions consisting essentially of glycosidic cleavage when 3-amino-4-hydroxybenzoic acid (AHB) with sodium is applied to glycopeptide and mucin samples.
Reviewer Comment: The experiments reported in this paper are designed to produce ISD fragment ions by use of a high laser power. From an analytical point of view, the initial MS spectrum should be as devoid of fragment ions as possible so that it reflects the composition of the sample. C-Type fragments, as are produced in these experiments, are isobaric with native glycans and their presence can confuse analytical results. Spectra should initially be recorded at a lower laser power to maximize the relative abundance of the molecular ions.
Author’s reply: As you have indicated, soft ionization methods such as MALDI should contain as few fragment ions as possible to reflect the composition of the sample. In fact, the synthetic glycopeptide sample used in this study was recorded and reported in the original literature (Ref. 23) using DHB as the matrix and a spectrum with minimal fragment ions. In addition, the mucin used in this study is a very large molecule with many mucin-type glycans, and ionizing it without fragmentation is itself difficult. Furthermore, even if soft ionization of mucin were possible, the O-glycan heterogeneity and stable isotope distribution, it is difficult to infer the structure from the generated parent ion. Therefore, the method of cleaving and separating glycans before measurement is generally used. This study aims to achieve ultra-rapid analysis by omitting this cleavage and separation process through in-source decay. Of course, the biggest problem is the information lost by this "omission" and the mixture of miscellaneous information that cannot be identified, and this does not negate the conventional precision analysis method. This manuscript proposes the use of a "crude but extremely rapid" glycan information acquisition method as a biotyping method, called glycotyping, by combining this rediscovered matrix and the in-source decay method.
Reviewer Comment: Analysis of the glycopeptide sample (a mixture of glycosylated cyclic peptides) is shown in Figure 2 with the four matrices tested. The most prominent molecular ions are produced from DHB, not the AHB matrix. AHB also produces more prominent fragment ions, including C-type ions which could be mistaken for additional compounds.
Author’s reply: Yes, the most prominent molecular ions from DHB is exactly why AHB is rarely used for analysis of carbohydrates and glycoconjugate anymore. Also, as you pointed out, the C-type fragment ions need to be handled with care because the free glycans mix to produce exactly the same ions. In order to rigorously verify this point, this study utilizes an oxime labeling method with benzyloxyamine (BOA), which takes advantage of the reactivity of the reducing end characteristic of free glycans to eliminate the possibility of the presence of free glycans.
Reviewer Comment: Figure 3d (spectrum of Gal-GalNAc) – This spectrum contains several unlabelled ions (e.g. m/z 394) not present in the other spectra in this figure. What are they?
Author’s reply: Unlabeled ions (e.g. m/z 394) are matrix-derived ions. In fact, this signal is observed in the spectra with and without the sample shown in Figure S5.
Reviewer Comment: The ion peaks in this spectrum are sharper than those from the other matrices. Have the authors ruled out the possibility that this effect could be caused by the use of different focusing conditions of the instrument?
Author’s reply: The sharpening of the molecular ion peak when using the AHB/Na matrix is expected to be the result of the overall effect of "In addition to glycan-selective ionization efficiency, nucleophilicity, the suppression of total ion production during intense laser irradiation, and morphological homogeneity" as described in the text. All experiments described in this study were performed using the instrumentation described in Section 3.4, "MALDI-TOF and TOF/TOF MS," and no adjustments were made among the experiments (spectra). Peak widths tend to become wider as the amount of ions handled by the mass spectrometer increases. In particular, since the instrument used in this study is not equipped with an ion trap mechanism, the ion width becomes wider when the ionization efficiency is high and the amount of foreign peaks derived from the matrix itself is overwhelmingly large, such as in the case of DHB and DAN. In addition, matrices that form giant crystals, such as DHB, tend to give broad peaks because the ion desorption direction becomes inhomogeneous. The morphology of the matrix is shown in Figure 3e to discuss this point.
Reviewer Comment: Figure 3 and later – The symbols used for the glycans should be defined. Do the angles connecting the symbols represent linkage?
Author’s reply: The symbols used for glycans are defined in Figures 3-6.
Reviewer Comment: Figure 3(e) The photographs show the MALDI targets produced by each matrix but there is no discussion in the text.
Author’s reply: As you indicated, the discussion of Figure 3(e) did not match the quoted portion of the text, so we have changed the location of the Figure 3(e) citation in the text, as noted earlier, as follows.
"In addition to glycan-selective ionization efficiency [12], nucleophilicity, the suppression of total ion production during intense laser irradiation, and morphological homogeneity (Fig. 3e and Fig. S1) [19, 20] are expected to contribute to the high resolution and high S/N ratio of O- glycan-derived ISD ions produced when using this AHB/Na matrix."
Matrix’s morphological homogeneity (planarity) contributes to narrow peak widths. Figure 3(e) shows the morphology of each matrix, which we believe is an important factor in forming the characteristics of the AHB/Na matrix. As shown in Figure 3 (e), matrix morphology is an important factor affecting peak line widths in "non-ion trap" MALDI-TOFMS, but as noted above, the low ionization efficiency and high ionization selectivity of AHB/Na under strong laser irradiation conditions may also be a significant factor.
Reviewer Comment: The analysis of the mucin sample presents a similar problem to that of the glycopeptide. Here, only fragment ions and no molecular ions are observed and so it is not possible to differentiate ions corresponding to intact glycans from C fragments produced from the larger glycans. Thus, the composition of the sample cannot be deduced from the spectrum. Each ion is then fragmented but only B and C glycosidic fragments appear to be formed. These ions are insufficient to determine the detailed structure of the ions (i.e. no cross-ring cleavages are reported). Yet, Figure 4 shows detailed structures, including the linkage between the monosaccharide constituents. How were the structures of these ions obtained? Many of the ions shown in the fragmentation spectra are unlabelled. Can the authors comment on the nature of these ions?
Author’s reply: As you point out, the O-glycan signals obtained in this study are limited compared to O-glycans after various pretreatments and ESI-MS, which can perform advanced MSn analysis. However, we believe that this study is the first positive example of obtaining "substantial" O-glycan information from a mucin sample without any pretreatment. Therefore, this study is about a "glycotyping" method with the greatest emphasis on rapidity, and as you pointed out, there is a lot of missing information to call it a "glycomics" method. However, we believe that this does not detract from the merits of this study.
Reviewer Comment: Figure 5 - the spectrum (b) contains many prominent ions not present in spectrum (a) even though the ions selected for fragmentation are nominally identical. Can the authors comment on the nature of these additional ions?
Author’s reply: Comparing Figure 3d with Figure 5, the cleanliness of the parent ions subjected to TOF/TOF analysis is quite different. Therefore, Figure 5b contains more molecular ions than Figure 5a because different molecular ions (noise ions) were included during parent ion selection.
Reviewer Comment: Supplementary figures S5, S6: Spectra show, in panels (b) contain ions which appear to be those from the matrix and to be present at abundancies similar to those from the sample. Surely, this is a major disadvantage?
Author’s reply: Compared to the DAN, DHB, and DAN/DHB/Na matrices shown in the capture figures S2, S3, and S4, the advantage of the AHB/Na matrix shown in the capture figures S5 and S6 is obvious. The matrix-derived signal in this region is an intrinsic drawback of the MALDI method with organic matrices, but the advantage of the AHB/Na minimizes this drawback of the MALDI method.
Reviewer Comment: Figures S8, S9 – most of the ions are unlabelled and many appear to be from the matrix. The labelled glycan-derived ions appear to be of very low abundance. Figures S10b to S19b – There is no indication as to what the ions represent.
Author’s reply: As you pointed out, most of the signals in Figures S8 and S9 are matrix-derived ions. This is a common problem with MALDI methods using organic matrices, as explained in the introduction and shown in Figures S2 to S4. This study describes data showing the advantages of AHB/Na over other organic matrices, but it is not intended to completely overcome this common problem with organic matrices.
Reviewer Comment:
References
Some points need attention:
Some journal titles are abbreviated, others are not.
Reference 9 – Normal text should be used for the article title, not upper case.
Reference 15 - The volume and page numbers are missing.
Reference 16 – The volume number is missing.
Author’s reply: Thank you for your careful review and pointing out the cited references. All have been corrected.
Reviewer Comment: In conclusion, the problem with this work is that there is no way to determine the glycan composition of the mucin sample because the liberated glycans have similar masses to C-type fragments that can be formed from many of the glycans.
Author’s reply: Because this study is focused on rapidity, we have not been able to fully overcome the problems you have pointed out. However, we have confirmed the identification with free glycans by validation using the BOA/DHB/Na matrix.
Reviewer Comment: The authors should investigate conditions for producing molecular ions from the glycans in the mucin sample other than by the production of ISD fragments. Possibly this will mean chemically liberating the glycans.
Author’s reply: Reference [28] shows a rapid MALDI-TOFMS method without generating ISD fragments after free glycans are chemically generated. In addition to this conventional approach, this study explores an approach using ISD fragmentation.
Reviewer Comment: Then the AHB matrix should be compared with the other matrices for their ability to produce molecular ion. Following this, fragmentation can be performed on the molecular ions known to be from native glycans, possibly by a method that produces cross-ring fragments in addition to the reported glycosidic fragments.
Author’s reply: References [28] and [29] show the results of analysis of porcine stomach mucin (PSM) O-glycans after they were made free. Comparison with the results of these references reveals the advantages and disadvantages of the studies we have described in this manuscript. In addition, as described in the introduction, the analysis of free glycans released from proteins by AHB matrix was reported in 1991 in Ref. [12] as the world first example of MALDI-TOFMS analysis of free glycan.
Reviewer 4 Report
Comments and Suggestions for Authors
The authors present an excellne transcript on the use of novel matrix for matrix-assisted laser desorption/ionization mass spectrometry (MALDI-MS). as well as the companion with other currently employed matrixes. In particular, the efficiency and sugar-selectivity of the ionization ability of AHB is demonstrated. Interestingly, the addition of sodium improves the properties of AHB. As application, the internal structure of different O-glycans from an intact mucin mixture is elucidated. Excellent from all perspectives
Author Response
Thank you for your positive feedback. We will continue to explore more practical matrices for medical applications of this discovery.